

# The influence of music teaching appreciation on the mental health of college students based on multimedia data analysis

Qiangwei Shen

School of Foreign Languages, Xinyang University, Xinyang, Henan, China

## ABSTRACT

The mental health problem of college students has gradually become the focus of people's attention. The music appreciation course in university is a very effective approach of psychological counseling, and it is urgent to explore the role of music appreciation in psychological adjustment. Therefore, we propose an emotion classification model based on particle swarm optimization (PSO) to study the effect of inter active music appreciation teaching on the mental health of college students. We first extract musical features as input. Then, the extracted music appreciation features generate subtitles of music information. Finally, we weight the above features, input them into the network, modify the network through particle swarm optimization, and output the emotional class of music. The experimental results show that the music emotion classification model has a high classification accuracy of 82.6%, and can obtain the emotional categories included in interactive music appreciation, which is helpful to guide the mental health of college students in music appreciation teaching.

## INTRODUCTION

With the emergence of more and more psychological problems in college students, society urgently needs proper psychological adjustment for students. Interactive music appreciation teaching is very influential among many adjusting methods. Based on the content of traditional narration education, multimedia adds new elements to make music education intuitive and vivid, significantly different from the conventional single narration way. Through multimedia, various music education situations are constructed, conducive to the interaction between teachers and students and enabling students to participate in music learning actively. At the same time, multimedia can centrally acquire and save information, integrate all kinds of resources, and present it in the form of data, which strengthens students' impression of knowledge points. Through multimedia, music education content and materials can be rearranged, avoiding the repetition and monotonicity of traditional education mode.

Corresponding author
Qiangwei Shen,
sqw_1234567@163.com

As a medium of emotion expression and emotion regulation, music can affect people's emotional state and mental health. Interactive music appreciation teaching can provide a richer and more personalized music experience, help college students better understand and express their emotions, and thus enhance their mental health. Proper music appreciation and participation in interactive music instruction can be effective ways to relieve stress and anxiety. The pleasure and relaxation effect of music can help to reduce the psychological burden and improve the happiness of students. Music has a positive effect on learning and memory. Through interactive music appreciation teaching, students can learn in a more fun and enjoyable atmosphere, increasing engagement and motivation in learning, and thereby enhancing learning outcomes. By studying the influence of interactive music appreciation teaching on the mental health of college students, we can better understand the role of music in education, and provide useful enlightenment and guidance for educational institutions and educators. Meanwhile, the teachers should concentrate on cultivating students' music participation consciousness, thinking ability, and language expression ability to improve students' music appreciation ability constantly. However, we need to choose the right music. Many kinds of music for college students trap into different psychological problems. Therefore, we must propose an emotion classification model in interactive music appreciation teaching.

In recent years, convolutional neural network (CNN) has become a hot basic structure with strong learning ability and has proven effective in many classification tasks. With the complete knowledge of CNN, *Costa, Oliveira & Silla Jr (2017)* designed a hybrid model to fuse CNN features and manual extraction, which improves the accuracy of music classification while comparing the results with audio features and SVM classifiers for music genre classification tasks. *Li, Chan & Chun (2010)* regard CNN as the feature-extracting tool and apply the majority voting mechanism to select the extracted features, which can achieve excellent performance on the GTZAN dataset. *Lee et al. (2018)* proposed a novel CNN using typical frame structure features. They also improve training speed by reducing the music signal frequency, which can shorten the training time.

Furthermore, they apply the migration study to extend the model and make the sample layer CNN visible by learning each layer filter, which can identify the hierarchical learning features. *Song et al. (2018)* extract the scattering transform elements and propose a new deep recurrent neural network (RNN) automatic labeling algorithm. The scattering features can realize the function of changing the spectrum for RNN with gate recurrent unit (GRU). Moreover, this transformation method is more efficient than MFCC and Mel spectrogram. Recently, attention mechanisms have been an effective tool for modeling the sequence in CV or NLP tasks. Among them, the self-attention mechanism associates different positions to compute their features within the same series, which are successfully utilized in a lot of research, such as text processing, artificial reading, *etc*. Subsequently, *Zhuang, Chen & Zheng (2020)* proposes a classifier based on a Transformer, which avoids recursion and convolutional structures to build the parallel mode that can cost less computing power than former methods with the result of improvement for music genre classification. The model is constructed by self-attention to analyze global relations and to build the connections between frames and voice information.

Music therapy is a method based on the theory of psychological diagnosis and treatment (*Lam et al., 2020*). Through the physiological and psychological reactions produced by music, with the help of music physiotherapists, using established musical interaction, and relying on music perception, patients can regain confidence and obtain healthy psychology. To build the model for the emotion with higher classifying accuracy for music appreciation, we analyze the defects of the existing classification of emotion methods in music appreciation. Through investigation and analysis, we can find that the deep network constructed the current optimal sentiment classification models in music appreciation (*Brancatisano, Baird & Thompson, 2020*; *Stegemann et al., 2019*). However, these models of emotion classification based on neural networks, whose error rate is higher, have low efficiency and may be trapped by the issue of local minimum with increasing appreciation of music data.

Therefore, we should re-structure the deep network and propose a model optimized by adopting a particle swarm to improve the emotion classification deep network for music appreciation, designed to speed up the optimization ability of neural network and increase the accuracy of classifying emotion in music appreciation. We first extract the music features as the input to construct the emotion classification model for music appreciation based on particle swarm optimization (PSO). Then, we generate captions for music information *via* the extracted features of music appreciation. Finally, we identify the emotion types in music appreciation. In this way, it is helpful to analyze the mental health problems of college students and put forward a plan to alleviate the psychological issues. Main contributions are as follows:

1. We extract short-term energy features, time-domain variance features, and other music appreciation features to construct the music appreciation classification model.
2. We apply the particle swarm optimization to boost the neural network, which can help us achieve the optimal results.

## RELATED WORK

Music therapy is widely accepted as an effective tool to relieve negative emotions. But traditional music therapy has the following problems: first, it is not clear the patient's emotional orientation; Second, different people have different feelings about the same concert. For students, it is difficult for music teachers to grasp the emotional feedback of each person accurately. Third, due to the numerous and miscellaneous music, it is difficult to simplify and improve the effect in a limited time. *Witteveen, Pradhapan & Mihajlović (2019)* found that β-level activity increased in high-pressure subjects. *Al-Ezzi et al. (2020)* found that SW/FW (slow/fast wave) negatively correlates with anxiety. Bos uses Fourier transform to obtain the energy ratio of EEG signals in different frequency bands, which can be used as a characteristic to characterize emotions. *Vujic et al. (2020)* found that by analyzing the EEG and barycenter power of the alpha band, Chinese classical music, Chinese rock music, Chinese pop music, and Banderi music were the most effective in alleviating sadness. In contrast, Chinese rock music had the worst effect. Therefore, the reasonable classification of music emotion is helpful to accurately grasp the direction of treating people's psychological problems.

*Li & Bin (2019)* apply the long short-term memory (LSTM) (*Hochreiter & Schmidhuber, 1997*) network in music genre classification and extract Mel frequency cepstral coefficient, spectral centroid, and spectral contrast features in the GTZAN dataset, who performs feature training *via* LSTM network. *He et al. (2022)* adopts the cepstrum coefficient to extract the MFCC feature matrix of audio. He also uses its eigenvalues as the input of CNN to train the music signal, which verified the effectiveness of CNN classification. *Bakhtyari, Davoudi & Mirzaei (2022)* propose a bidirectional recurrent neural network using STFT spectrogram as input features, which combines serial and parallel attention mechanisms to evaluate the performance of the model above on the GTZAN dataset and Extended Ballroom in experiments. *Liu (2021)* tries to solve the problem of deep learning in music tasks with low accuracy and proposes the feature index system for music according to the basic theory of acoustics and signaling in the view of the perspective of music features, namely music genres classification based on the algorithm of adaptive particle swarm optimization neural network.

Nowadays, machine learning and deep learning (*Pandeya, Bhattarai & Lee, 2021*; *Hizlisoy, Yildirim & Tufekci, 2021*) are combined to classify music. Deep learning adopts supervised or semi-supervised efficient algorithms to reduce the error of artificial features extracted *via* traditional machine learning algorithms, improving music classification (*Mengqing, 2021*). However, the cost of deep learning is high, and the model design is complex, which needs to be further enhanced by combining the music feature information to improve the overall performance of classification.

# EMOTION CLASSIFICATION NETWORK BASED ON PSO FOR APPRECIATIVE MUSIC TEACHING

## The framework

The steps of interactive music emotion classification based on deep network optimized by PSO are presented as follows (*Amjad, Khan & Chang, 2021*; *Modran et al., 2023*):

1) Extract different categories of features from interactive music.

2) We apply the grey relational analysis method to calculate the contribution of different interactive music features. Then, we use the assistance to represent the importance of interactive music features for emotion classification.

3) The PSO improves the deep network to obtain the best neural network. The interactive music emotion classification results are obtained through the web.

The framework of the interactive music sentiment classification model is shown in Fig. 1.

As shown in Fig. 1, we first collect different classes of ecological appreciation music information and denoise the appreciation music data before re-constructing the model of classifying emotion for music therapy. Then, we conduct framing and endpoint detecting to achieve appreciated music information for denoised appreciation music. Subsequently, we extract the features of Time-Domain, the characteristics of variance in Frequency-Domain, and the energy features with short-term from the detected music information to combine them into a part. Next, we apply the grey relational analysis method to determine the contribution of the three features for music classification and weigh the features above

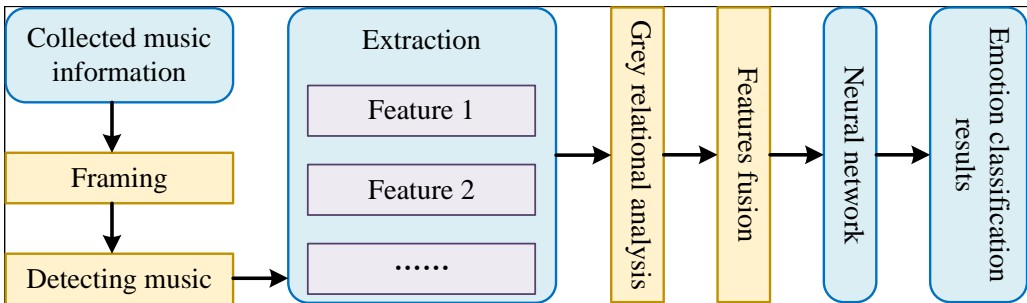

**Figure 1  The framework of interactive music emotion classification model.**

to fuse multiple features. We input the weighted three features into the deep network fine-tuned by PSO. Finally, we can obtain the classification results of emotion in music appreciation by the deep network fine-tuned by PSO.

## Multi-feature fusion of music
### Extraction of music features
We must extract short-term energy features, time-domain variance features, and other music appreciation features to construct the music appreciation classification model for the diversity of music therapy features (*Modran et al., 2023*). The specific extraction process is as follows:

1) Short-term energy features of appreciation music. There is a big difference between the energy of ordinary sound and appreciation music. Compared with ordinary sound, appreciation music has a higher energy value. Therefore, we can extract the energy features of short-term from each frame in appreciated music. Assume that $E_m$ represents the energy of the music appreciation signal $\{y(m)\}$. The short-term energy of the music appreciation signal $\{y(m)\}$ is calculated as follows:

$$E_m = \sum_{-\infty}^{+\infty} \left[ y(n) \cdot c(n-m) \right]^2 \tag{1}$$

where $c(m)$ refers to the window function. We regard N as the duration of the appreciated music. Therefore, $E_m$ can be presented as follows:

$$E_m = \sum_{n=0}^{N-1} \left[ y(n) \cdot c(n-m) \right]^2 \tag{2}$$

2) The variance features Time-Domain in appreciated music. The following formula can express any frame of appreciated music:

$$Y_t(n) = \left\{ y_t(n,1), y_t(n,2), \ldots, y_t(n,N) \right\} \tag{3}$$

where $Y_t(n)$ represents a frame of appreciation music. We apply the wavelet transform smoothing operation for $Y_t(n)$ and achieve the processed frame $\hat{Y}_t(n)$ of appreciation

music. At this time, a frame of appreciation music after processing can be described by the following formula; the specific formula is as follows:

$$\hat{Y}_t(n) = \left\{ \hat{y}_t(n,1), \hat{y}_t(n,2), \ldots, \hat{y}_t(n,N) \right\} \tag{4}$$

Combined with formula (4), we calculate the mean $E_t(n)$ and variance $D_t(n)$ of the time domain of music appreciation. The formula for calculating the mean and variance is as follows:

$$E_t(n) = \frac{1}{N} \sum_{i=1}^{N} \hat{y}_t(n,i) \tag{5}$$

$$D_t(n) = \frac{1}{N} \sum_{i=1}^{N} \left[ \hat{y}_t(n,i) - E_t(n) \right]^2 \tag{6}$$

3) The variance features of Frequency-Domain in appreciated music. Any frame $Y_f(n)$ of appreciated music is obtained from the frame $Y_i(n)$ by Fourier transform. Then the description of the specific frame of music appreciation after transformation is represented in the following formula:

$$Y_f(n) = \left\{ y_f(n,1), y_f(n,2), \ldots, y_f(n,N) \right\} \tag{7}$$

We adopt the wavelet transform smoothing operation $Y_f(n) = \frac{1}{N} \sum_{i=1}^{n} \left[ \hat{y}_f(n,i) - E_f(n) \right]^2$ to obtain $\hat{Y}_f(n)$, whose expression is as follows:

$$\hat{Y}_f(n) = \left\{ \hat{y}_f(n,1), \hat{y}_f(n,2), \ldots, \hat{y}_f(n,N) \right\} \tag{8}$$

We associate Eq. (8) to calculate the mean $E_f(n)$ and variance $D_f(n)$ of music appreciation in the frequency domain. The specific calculation formula is as follows:

$$E_f(n) = \frac{1}{N} \sum_{i=1}^{N} \hat{y}_f(n,i) \tag{9}$$

$$D_f(n) = \frac{1}{N} \sum_{i=1}^{N} \left[ \hat{y}_f(n,i) - E_f(n) \right]^2 \tag{10}$$

### Feature fusion

1) We assume $Y_i = \left( y_i(1), y_i(2), \ldots, y_i(m) \right), i = 1, 2, \ldots, m$ to represent the feature vector composed of the short-time energy feature, the variance features of Time-Domain, and the variance features of Frequency-Domain in the appreciated music.

2) We apply the preprocessing $Y_i = \left( y_i(1), y_i(2), \ldots, y_i(m) \right)$ To set the reference object, the randomly selected feature vectors. We regard the selected feature vectors

as $Y_0' = (y_0'(1), y_0'(2), \ldots, y_0'(m),)$ And combine the selected feature vectors and other groups of feature vectors as a pair of feature vectors.

3) We apply $\Delta_i(k) = |y_0'(k) - y_i'(k)|$ To compute the maximum and minimum deviation of a feature vector group. The detailed calculation process of the maximum deviation and minimum deviation is as follows:

$$\begin{cases} P_1 = \max\max \Delta_i(k) \\ P_2 = \min\min \Delta_i(k) \end{cases} \tag{11}$$

4) We calculate the grey correlation coefficient of music appreciation features. The formula is:

$$g_i(k) = \frac{n + \alpha M}{\Delta_i(k) + \alpha M} \tag{12}$$

where $\alpha$ represents the resolution coefficient,

We calculate the grey correlation degree of music appreciation features. The formula is as follows:

$$g_i = \sum_{k=1}^{n} w_i g_i(k) \tag{13}$$

where $w_i$ represents the weight. According to Equation (13), the contribution value of music appreciation features can be obtained. The contribution value can describe the influence degree of each feature on music appreciation.

Through the above methods, we can directly obtain the short-time energy feature, time domain variance feature and frequency domain variance, and realize the fusion of their features. Using the fusion feature, we adopt ResNet as the backbone of our method, with the help of ResNet's residual structure, to achieve the classification of musical emotions. However, considering the characteristics of feature fusion, we need to optimize the neural network structure.

## Network optimized by particle swarm

We optimize the deep network by the PSO to obtain the optimal neural network structure. Subsequently, we can achieve the classification results through the optimal neural network structure. Before we utilize the PSO to optimize neural networks, we should first analyze how the principle of the PSO algorithm improves the system of deep networks (*Das, Pattnaik & Padhy, 2014*). The following part is a detailed description of the direction and the process of the PSO algorithm to optimize the neural network.

In interactive music appreciation, emotion classification is an important task, which aims to understand and capture the emotions generated by listeners when they enjoy music. In order to effectively classify emotions, we modify the neural network to improve the classification performance. Considering that music feature is a multi-modal fusion feature, and PSO can quickly find the global optimal solution in the search space, the global search strategy helps to avoid falling into the local optimal solution, thus improving

the generalization performance of the network. In addition, the music feature is a high-dimensional feature, and PSO performs well in high-dimensional Spaces because it does not involve gradient information, but optimizes the objective function through a simple and easy-to-understand particle update mechanism

### PSO

The process of the PSO algorithm is as follows: we assume that the search space is L dimensions, in which the groups occupy the K particles. The positions of the j-th particle are expressed as $X_j = (x_{j1}, x_{j2}, , \ldots, x_{jl})$. The corresponding optimal solution for the positions of the j-th particle is $P_j = (p_{j1}, p_{j2}, , \ldots, p_{jl})$. $p_y$ represents the optimal global individual. The speed for the positions of the j-th particle is expressed as vectors $V_j = (v_{j1}, v_{j2}, , \ldots, v_{jl})$. The position and velocity of each particle with the change of iteration can be represented as follows:

$$v_{jl}(t+1) = g \cdot v_{jl}(t) + z_1 \cdot rd \cdot (p_{jl}(t) - x_{jl}(t)) +$$
$$z_2 \cdot rd \cdot (p_y(t) - x_{jl}(t)) \tag{14}$$

$$x_{jl}(t+1) = x_{jl}(t) + v_{jl}(t+1) \tag{15}$$

where g represents the inertia factor, $z_1$ and $z_2$ denotes the acceleration factor and the acceleration factor is an average number. In addition, rd indicates a random value between $[0, 1]$ and t refers to the current iteration number. Since the velocity and initial position are randomly set, the position and velocity of the particle can be iterated through Eqs. (14) and (15) until the termination can be realized or the iteration reaches the ending. The formula of inertia weight in Eq. (14) is as follows:

$$g = \begin{cases} g_{max} - \dfrac{(g_{max} - g_{min})(s - s_{ave})}{s_{max} - s_{ave}}, & s \geq s_{ave} \\ g_{max} & s < s_{ave} \end{cases} \tag{16}$$

where s represents particle adaptation value, $s_{ave}$ and $s_{max}$ represent the average adaptation value of particles in each generation and the maximum adaptation value in the particle swarm, respectively.

### Steps of optimization

According to the principle of the particle swarm optimization algorithm analyzed above, we can achieve the optimal thresholds and weightings by training the weightings and thresholds of the deep network with the PSO. Therefore, we can construct the network by these optimal weights and thresholds. The optimization process can be as follows:

1) First, we normalize the training and testing samples and set up a three-layer neural network topology structure. Then, we regard dimension M of the weighted three vectors obtained in section 1.2 as the input nodes number $S_I$. Besides, we take many appreciative music classes $L_B$ to be the output nodes $S_I$ of network. The geometric average of input and output layer nodes is the number of the hidden node. The individual particles in the

population are represented as real number vectors obtained by encoding all the thresholds and connection weights between neurons.

2) Design the fitness function. The fitness function is the metric of evaluating the neural network to solve problems. We adopt the mean square error of the output value of our model to obtain the objective function. Furthermore, we treat the inverse of the objective function as the fitness function.

3) Initialize and set parameters. The parameters include the number of particles, initial position, and velocity. We control the initial position and velocity between [0,1] to train our model faster. We also set the acceleration factor and the maximum number of iterations simultaneously.

4) According to the input and output samples, we calculate the fitness function to obtain the value of individual particles using Eq. (17). At the same time, we regard the historical best position as the best position of each particle to start the iteration for the particle.

5) We follow Eqs. (14)–(16) in the particle swarm algorithm to renew the velocity of particles and the position of particles. Then, we check whether the location of the particles and the momentum of the particle are trapped in the boundary. If particles have crossed the border, it is necessary to exclude the crossing particles and update the velocity of particles and the position of the particles again.

6) We calculate the fitness function to obtain a value of the updated particles and change the inertia weight according to the fitness value. Then, we search for the best position of the particle and check whether it satisfies the end judgment.

However, the performance of PSO is affected by the selection of the initial particle swarm, and local optimization is still a potential influence for complex non-convex problems. It is difficult to adjust parameters, such as inertia weight, individual learning factor and group learning factor, and the setting of these parameters may affect the performance of the algorithm. Meanwhile, compared with our proposed model, user survey and psychological test can directly obtain students' subjective feedback and psychological feelings, which involves higher labor and time costs. Our model can quickly process large amounts of data and automate sentiment classification, saving labor and time costs.

## EXPERIMENTS

### Dataset

To experiment with classifying emotion in music therapy, we collect the different types of music, such as ensemble music, orchestral music, silk and bamboo music, piano music, pop music, and so on., to construct our dataset. The training set contains 50,000 samples, and the test set includes 5,000 pieces.

Each track in the dataset was sourced from publicly available music databases and labeled as belonging to a different emotional category. The characteristics of music emotion classification tasks usually involve the processing of audio signals and the extraction of music metadata. Accuracy is an important metric in the music emotion classification task because we want the model to accurately predict the listener's emotional response to the music. The confusion matrix provides more comprehensive performance information,

**Table 1  Training setting.**

| Hyper parameters | The values |
|---|---|
| Learning rate | 0.0002 |
| Dropout | 0.4 |
| Model optimizer | SDG |
| Number of Epoch | 150 |
| Batch size | 32 |

**Table 2  Different types of music.**

| Labels | Types name | Labels | Types name |
|---|---|---|---|
| 1 | Ensemble music | 6 | Vocal music |
| 2 | Orchestral music | 7 | Military music |
| 3 | Silk & bamboo music | 8 | Symphony |
| 4 | Piano music | 9 | Instrumental music |
| 5 | Chamber music | 10 | Percussion music |

helping us understand how the classifier performs on different emotional categories, as well as the types of classification errors that may exist.

## Implement details

We conduct our experiments with the i7-12900k Cpu, 3080ti Gpu, and the PyTorch deep learning framework. The training settings of the classification model of emotion in interactive music appreciation teaching are presented in Table 1. In addition, we set the Swarm Size to 200, set the Inertia Weight to 0.9, set the Cognitive Learning Factor to 1.5, and set the Social Learning Factor to 1.2 to construct the PSO.

We apply the accuracy as the evaluating metric, which is presented as follows:

$$Accuracy = \frac{I_c}{I_{total}} \tag{17}$$

where $I_c$ refers to the accounts of correct classification samples and $I_{total}$ relates to the counts for the whole samples.

## Results and Comparison

### Classes for the emotion of music therapy

Before conducting experiments to compare with other methods, we should divide all music samples into different types to prepare for the further analysis of emotion, as shown in Table 2. Furthermore, we artificially set the music of emotion to 10 other emotions: anxiety, fear, depression, euphoria, melancholy, happiness, sadness, peace, and irritability.

In addition, we analyze the distribution and statistics of the entire dataset; the results are presented in Fig. 2. According to this, we can demonstrate that the dataset we collect is excellent for meeting the demand for emotion classification. Then, we count the distribution of the ten emotions in our dataset. The distribution and statistics are presented in Figs. 3

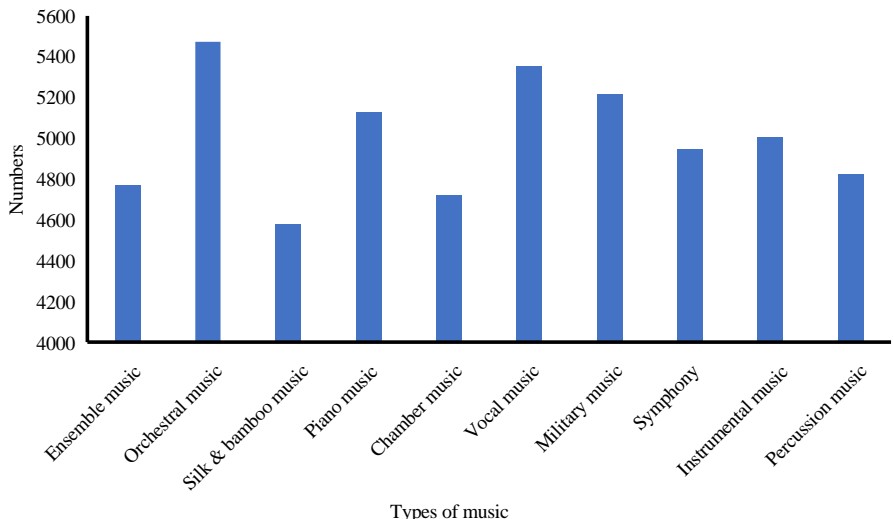

**Figure 2** **The distribution of various music.**

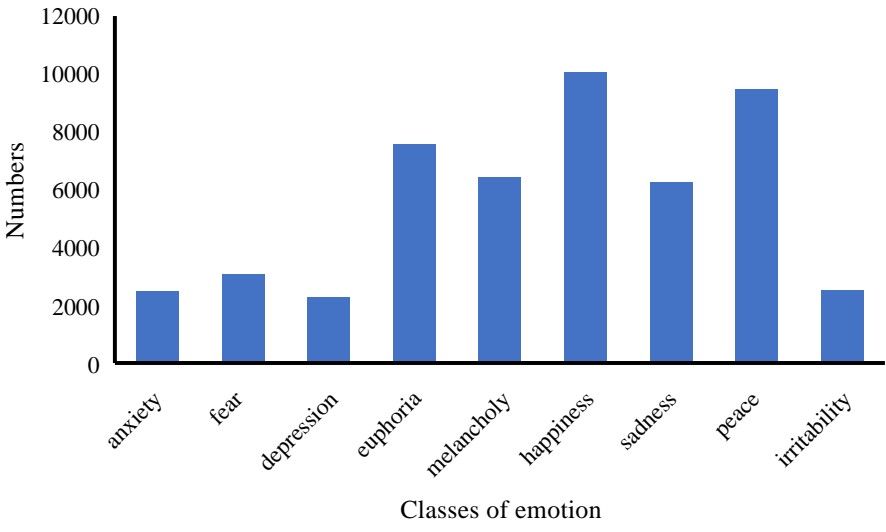

**Figure 3** **The distribution of emotion classes.**

and 4. It can be seen that the feelings of happiness, euphoria, and peace occupy the high ratio, which are the primary emotions.

### Comparison with other methods

To evaluate the classification results of our model in this paper, we adopt the interactive music appreciation pitch and length to represent the frequency features of music appreciation. At the same time, we apply the reverberation time and tone to describe the time characteristics of music appreciation. Besides, the mean square of short-term

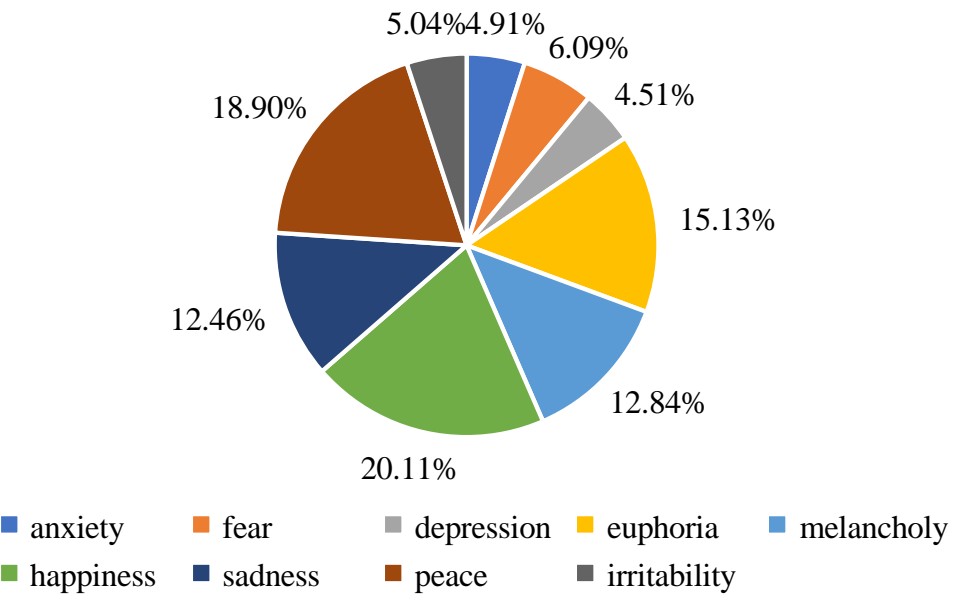

**Figure 4** The statistics of emotion classes.

**Table 3** Results about classification of emotion between different types of music.

| Music types | Pitch | Length | Tone | Reverberation time | Accuracy (%) |
|---|---|---|---|---|---|
| Ensemble | 5.11 | 60.4 | D | 0.26 | 74.1 |
| Orchestral | 5.13 | 61.2 | Bass Flute | 0.09 | 85.6 |
| Silk & bamboo | 4.65 | 45.9 | D | 0.26 | 77.0 |
| Piano | 4.26 | 46.3 | C | 0.18 | 82.5 |
| Military | 5.21 | 53.8 | Bass Flute | 0.08 | 84.3 |
| Chamber | 4.36 | 44.9 | D | 0.16 | 86.8 |
| Instrumental | 5.19 | 53.9 | D | 0.13 | 88.8 |
| Vocal | 4.18 | 46.2 | G | 0.26 | 80.3 |
| Symphony | 4.07 | 49.1 | D | 0.09 | 82.6 |
| Percussion | 5.32 | 57.6 | G | 0.06 | 84.2 |
| Average | – | – | – | – | 82.6 |

energy in music represents the energy features of music appreciation. We use the music appreciation emotion classification model to output the 10 emotion classes of music appreciation. The performance of our model is presented in Table 3.

As shown in Table 3, we can demonstrate that our model achieves excellent classification performance on the emotion of interactive music appreciation teaching. Our model can classify the emotion in the music of Ensemble, Orchestral Silk & bamboo, Piano, Chamber, Vocal, Military, Symphony, Instrumental, and Percussion with accuracy of 74.1%, 85.6%, 77.0%, 82.5%, 84.3%, 86.8%, 88.8%, 80.3%, 82.6%, and 84.2%, respectively, whose average score of accuracy can reach 82.6%. By comparing the classification results with the actual emotion categories of music, we can find that the results of the emotion classification

model for appreciated music are the same as the ground truth of appreciated music, which indicates that the emotion classification model of interactive music appreciation teaching is highly accurate. We can conclude that our model can effectively classify the emotion in the interactive music appreciation teaching to adjust the psychology of college students. Since we use three completely different features to predict emotion in music, our model is able to guarantee the prediction effect in different data sets and subjects, and our prediction accuracy can verify this.

Music appreciation teaching can provide students with rich and colorful music experience and help them better understand the language and expression of music. By learning about different types of music and emotional categories, students can develop an appreciation for music and an aesthetic awareness. Music appreciation teaching can also enhance students' emotional expression ability, improve emotional cognition and emotional management ability, and play a positive role in relieving stress and emotional regulation.

The significance of accurate predicting emotional categories for teaching strategies and mental health. Accurately predicting students' emotional categories can help teachers better understand students' emotional states and emotional needs. According to the predicted results of different emotional categories, teachers can formulate corresponding teaching strategies to promote students' positive mental health. For students to show positive emotions, teachers can use more active and pleasant music to teach, stimulate students' learning interest and participation. For students who show negative emotions, teachers can adopt a more caring, supportive and understanding attitude, provide positive music to relieve them, encourage students to talk about their emotions and seek professional help.

In addition, the accurate prediction of emotional categories can also be used to evaluate teaching effectiveness and mental health status, providing schools with important student emotional data and mental health feedback. By obtaining the information of students' emotional state in time, schools can take more effective mental health intervention measures, provide necessary psychological counseling and support services, and promote students' all-round development and mental health.

## CONCLUSION

To study how to regulate the psychology of college students through interactive music appreciation teaching, we propose a method to classify the emotions of interactive music to achieve the influence of music on people's emotions. First, we extract different categories of speech information from the music and de-noise it. Then, we perform frame splitting and endpoint detection on the music we enjoy. Next, time domain features, frequency domain variance features and short-term energy features are extracted to improve performance. Finally, the contribution of features is calculated by using grey relational analysis and the features are weighted. Subsequently, we feed these features into a deep network optimized by the PSO. The experimental results show that the model has a good effect of music therapy emotion classification, the accuracy rate can reach 82.6%, and further regulate the psychology of college students.

## ACKNOWLEDGEMENTS

I would like to thank my college for its support of my work.

### Funding
The authors received no funding for this work.

### Competing Interests
The authors declare there are no competing interests.

### Author Contributions
- Qiangwei Shen conceived and designed the experiments, performed the experiments, analyzed the data, performed the computation work, prepared figures and/or tables, authored or reviewed drafts of the article, and approved the final draft.

### Data Availability
The code is available in the Supplemental File.

The data is available at Zenodo: Yagya Raj Pandeya. (2021). Deep Learning-Based Multimodal Emotion Classification for Music Videos (Version V2) [Data set]. Zenodo. https://doi.org/10.5281/zenodo.4542796.

### Supplemental Information
Supplemental information for this article can be found online at http://dx.doi.org/10.7717/peerj-cs.1589#supplemental-information.

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
