# Peer review of "The influence of music teaching appreciation on the mental health of college students based on multimedia data analysis"

_PeerJ Computer Science, doi:10.7717/peerj-cs.1589_

## Round 0.1 · original submission · Major Revisions

Dear authors
Thanks for your submission, your article has been reviewed by the experts in the field and you will see that they have some valid comments for the improvements of the article, therefore you are requested to please improve in the light of these comments.

Please also improve the language of the manuscript professionally,

Also, improve the methodology in detail.

**Language Note:** The Academic Editor has identified that the English language must be improved. PeerJ can provide language editing services - please contact us at copyediting@peerj.com for pricing (be sure to provide your manuscript number and title). Alternatively, you should make your own arrangements to improve the language quality and provide details in your response letter. – PeerJ Staff

Reviewer 1 ·

Basic reporting

I have reviewed the manuscript of your research and identified several suggestions for revision to enhance its content. Here are the proposed amendments:

• Clarify the research objective: Clearly state the main objective of the research in the abstract. Specify whether the aim is to evaluate the impact of interactive music appreciation teaching on college students' mental health or to develop a music appreciation emotion classification model.

• Add a research gap: Introduce the existing gaps in the literature that your research aims to address. Explain why the proposed study is necessary and how it contributes to the current knowledge on the topic.

• Elaborate on the methodology: Expand on the details of the proposed music appreciation emotion classification model. Explain how the musical features are extracted, the process of generating music information subtitles, and the specific network architecture being used. This will provide a clearer understanding of the technical aspects of your study.

• Justify the use of particle swarm optimization: Elaborate on why particle swarm optimization (PSO) is chosen as the optimization technique for modifying the network. Discuss its advantages in the context of emotion classification in interactive music appreciation. Additionally, consider mentioning any alternative optimization techniques that were considered and why PSO was preferred.

• Describe the dataset and evaluation metrics: Provide information on the dataset used for training and evaluating the music emotion classification model. Specify the size, source, and characteristics of the dataset. Additionally, mention the evaluation metrics used to measure the classification accuracy and explain why they are appropriate for the study.

• Include details on the experimental setup: Describe the experimental setup used to validate the proposed model. Specify the parameters used in the PSO algorithm, the number of iterations, and any other relevant details. This information will help readers understand the reproducibility and reliability of the experimental results.

• Discuss the limitations: Acknowledge the limitations of your study, such as potential biases, generalizability constraints, or assumptions made during the research process. Addressing the limitations will demonstrate a critical evaluation of your work and provide avenues for future research.

• Highlight practical implications: Discuss the practical implications of your findings for music appreciation teaching and college students' mental health. Explain how the accurate prediction of emotion categories can guide instructors in creating appropriate teaching strategies and fostering positive psychological health among students.

• Conclude with key findings: Summarize the main findings of your study, highlighting the high classification accuracy achieved by the music emotion classification model. Emphasize the potential impact of these findings on music appreciation teaching and college students' psychological well-being.

• Rewrite the Abstract and Conclusion sections as per the standard format including the main results.

By incorporating these amendments into your paper, you will provide a more comprehensive and informative paper, attracting readers and increasing the clarity and significance of your research. Remember to ensure that the full paper covers these aspects in detail as well.

Experimental design

.

Validity of the findings

.

·

Basic reporting

To research on how to adjust the psychology of college students through interactive music appreciation teaching, the authors propose a method to classify the emotion of the interactive music to reflect the influence of the music on people’s feelings. This is an interesting study, but the manuscript needs to be refined.

(1)   Explicitly explain why the study on the impact of interactive music appreciation teaching on college student's mental health is important. Discuss the potential benefits of addressing this issue, such as improved student well-being, enhanced learning outcomes, and better overall educational experiences.

Experimental design

(2)   Justify the selection of specific musical features used as input for the emotion classification model. Explain how these features are related to emotional responses and why they are relevant for the study of interactive music appreciation.

(3)   Elaborate on the methodology or algorithms used to generate subtitles based on the extracted music appreciation features. Describe any linguistic or semantic analysis techniques employed and how they contribute to the overall understanding and interpretation of the music.

(4)   Provide more details on the network architecture used for emotion classification. Explain the specific type of neural network employed (e.g., convolutional neural network, recurrent neural network) and discuss any modifications or adaptations made to suit the task of music emotion classification.

(5)   Explain the procedure followed to train the emotion classification model. Specify the number of training iterations, the learning rate, and any regularization techniques utilized. Additionally, mention how the model's performance was evaluated during training (e.g., validation set, cross-validation).

Validity of the findings

(6)   Address the generalizability of the developed music emotion classification model. Explain whether the model is expected to perform well on different datasets or if its performance is limited to specific types of music or student populations. Consider discussing potential biases or limitations that may affect the model's general applicability.

(7)   Explore the possibility of using alternative evaluation methods, such as user surveys or psychological tests, to validate the impact of interactive music appreciation teaching on college student's mental health. Discuss the advantages and limitations of these methods compared to the proposed emotion classification model.

(8)   Conclude the abstract by highlighting potential avenues for future research in this field. Suggest possible extensions to the current study, such as exploring the effectiveness of different teaching approaches, investigating the role of individual differences in emotional responses to music, or examining the long-term effects of music appreciation on mental health outcomes.

---

## Round 0.2 · accepted · Accept

Based on the improvement and input from the experts I'm pleased to accept your article.

Thanks for your fine contribution to our esteemed journal

Reviewer 1 ·

Basic reporting

The authors have addressed all the changes.

Experimental design

The authors have addressed all the changes.

Validity of the findings

The authors have addressed all the changes.

Additional comments

The authors have addressed all the changes.

·

Basic reporting

The authors have addressed all the concerns.

Experimental design

The authors have addressed all the concerns.

Validity of the findings

The authors have addressed all the concerns.